# Comparison of Wavelet Artificial Neural Network, Wavelet Support Vector Machine, and Adaptive Neuro-Fuzzy Inference System Methods in Estimating Total Solar Radiation in Iraq

Wongchai Anupong [1,*], Muhsin Jaber Jweeg [2], Sameer Alani [3], Ibrahim H. Al-Kharsan [4], Aníbal Alviz-Meza [5,*] and Yulineth Cárdenas-Escrocia [6]

1 Department of Agricultural Economy and Development, Faculty of Agriculture, Chiang Mai University, Chiang Mai 50200, Thailand
2 College of Technical Engineering, Al-Farahidi University, Baghdad 10001, Iraq
3 The University of Mashreq, Baghdad 10001, Iraq
4 Computer Technical Engineering Department, College of Technical Engineering, The Islamic University, Najaf 54001, Iraq
5 Grupo de Investigación en Deterioro de Materiales, Transición Energética y Ciencia de datos DANT3, Facultad de Ingeniería y Urbanismo, Universidad Señor de Sipán, Km 5 Via Pimentel, Chiclayo 14001, Peru
6 GIOPEN, Energy Optimization Research Group, Energy Department, Universidad de la Costa (CUC), Cl. 58 ##55-66, Barranquilla 080016, Atlántico, Colombia
* Correspondence: anupong.w@cmu.ac.th (W.A.); alvizanibal@crece.uss.edu.pe (A.A.-M.)

**Abstract:** Estimating the amount of solar radiation is very important in evaluating the amount of energy that can be received from the sun for the construction of solar power plants. Using machine learning tools to estimate solar energy can be a helpful method. With a high number of sunny days, Iraq has a high potential for using solar energy. This study used the Wavelet Artificial Neural Network (WANN), Wavelet Support Vector Machine (WSVM), and Adaptive Neuro-Fuzzy Inference System (ANFIS) techniques to estimate solar energy at Wasit and Dhi Qar stations in Iraq. RMSE, EMA, $R^2$, and IA criteria were used to evaluate the performance of the techniques and compare the results with the actual measured value. The results showed that the WANN and WSVM methods had similar results in solar energy modeling. However, the results of the WANN technique were slightly better than the WSVM technique. In Wasit and Dhi Qar stations, the value of $R^2$ for the WANN and WSVM methods was 0.89 and 0.86, respectively. The value of $R^2$ in the WANN and WSVM methods in Wasit and Dhi Qar stations was 0.88 and 0.87, respectively. The ANFIS technique also obtained acceptable results. However, compared to the other two techniques, the ANFIS results were lower, and the $R^2$ value was 0.84 and 0.83 in Wasit and Dhi Qar stations, respectively.

**Keywords:** solar energy; WANN; WSVM; ANFIS



## 1. Introduction

Human life has always depended on how energy is used to perform activities [1,2]. Since the beginning of history, identifying energy sources and their optimal exploitation has been considered one of the crucial tools for the development and economic self-sufficiency of any country. Population increase and the expansion of industrial activities are two factors that led to an increasing trend in electricity consumption worldwide. The need for electricity increased from 14,000 TWh in 2000 to more than 25,000 TWh in 2019. Known sources of energy currently available to humanity include fossil fuels (coal, oil, and natural gas), wood, surface water flow such as rivers and waterfalls, wind, sea waves, geothermal, nuclear, and solar [1–3]. Providing the required energy with oil and other fossil fuels has caused adverse environmental consequences, especially damage to the climate, land, and wildlife. Due to the shortage in fossil fuel sources, the change in their cost, and environmental problems, cleaner energy technologies utilizing renewable energy sources such as wind and solar have

been developed recently [4,5]. According to the literature, solar energy is a clean, renewable, and endless resource that helps preserve valuable natural resources such as oil and gas for future generations. Of course, the energy of solar radiation reaching the earth is not the same all over the world, and it is at a maximum in the areas near the equator [6]. One of the most important applications of solar energy is converting solar energy into electrical energy with photovoltaic systems without moving mechanisms. We can use solar energy by using solar panels and unglazed transpired solar collectors (UTCs) for different purposes such as food processing, wastewater treatment, home heating, commercial heating, and institutional heating [7]. This advantage has made it possible to use this type of solar energy on small and large scales, from mill-watts to several mega-watts. Additionally, building photovoltaic systems is very easy compared to other types of renewable energy, such as wind and geothermal energy. For this reason, constructing photovoltaic solar power plants is suitable for providing sustainable energy worldwide. Besides this advantage, selecting the most suitable places for exploiting solar energy depends on technical and economic climatic factors and criteria, so that the more these factors are optimized, the more energy will be received, and the initial investment and operation costs will be lower [6,8]. In the studies related to the assessment of solar radiation intensity (Rs), it is impossible to include all the effective elements in the equations. As a result, only a limited number of climatic variables are used to assess the intensity of solar radiation by empirical and semi-empirical equations [9]. In recent years, researchers have focused their studies on using data-driven methods and machine learning to estimate meteorological variables. The results of these studies have shown that using these methods has advantages and disadvantages. The most important advantage of these methods is the description and revealing of complex relationships. Despite the advantages that data-driven and machine-learning methods have over experimental and semi-experimental methods, there are also disadvantages in these methods, including the accuracy of the results, which greatly depends on the size of the training set [10,11].

Researchers have shown that the usage of data-driven methods and machine learning in estimating the amount of solar energy have performed well and can be suitable tools for the initial evaluation of solar energy production on a site. In China, the ability of the support vector machine (SVM) method to model solar radiation was investigated. The results indicated that the SVM method was able to be successfully employed in solar radiation modeling [12,13]. In Nigeria, SVM and adaptive neuro-fuzzy inference system (ANFIS) methods were used to model solar radiation [14]. A study investigated the thermal performance of energy conversion systems using solar energy. In this research, different approaches for modeling, including artificial neural networks, were investigated. The results showed that artificial neural networks could achieve high accuracy [15]. The solar potential has been evaluated in India using neural networks and the GIS technique. Using the artificial neural network (ANN) model was suggested to solve complex problems [11]. In a study in Eritrea, the potential of solar energy was estimated using remote sensing tools and GIS. The results showed the proper performance of these tools [16]. Solar radiation in several stations was evaluated in Turkey using Resilient Propagation learning algorithms and the ANN technique. The outcomes demonstrated that the artificial neural network technique had better results than other algorithms [8]. In Singapore, solar radiation was modeled using the ANN technique with fuzzy logic preprocessing and a forward neural network with error back propagation. Researchers have stated that the reason for using fuzzy logic preprocessing was to improve the error correction coefficient to reduce the prediction error [9]. In Brazil, researchers evaluated direct radiation hourly and daily using two methods, ANN and SVM, with 13-year data. The results showed the positive performance of the two methods [17]. Additionally, using an ANN for predicting the potential of solar, water, and wind energy sources has been investigated, and researchers have identified neural networks as reliable models for investigating energy sources [18].

Considering the importance of the accurate estimation of solar radiation intensity in stations that cannot be measured, on the one hand, and the use of small variables in

experimental equations, on the other hand, it is necessary to investigate the non-linear relationships between meteorological variables affecting solar radiation intensity using data-driven and machine learning methods. In this research, based on the number of sunny days and the high potential of solar energy in Iraq, an estimate of the amount of solar energy (Rs) has been attempted using a wavelet artificial neural network (WANN), wavelet support vector machine (WSVM), and ANFIS methods. In this study, the prediction of the intensity of the sun's radiation has been made using the effect of wavelet on the study of ANN and SVM methods. The input data has been decomposed into a subseries of components by different wavelets at different levels. Then, the sub-series analyzed by the mother wavelets were entered into the ANN and SVM methods to increase the accuracy of forecasting the intensity of solar radiation.

As discussed in the literature review, the previous research mainly investigated two machine learning methods. In this research, three methods were used to make a more comprehensive comparison between machine learning methods. Additionally, the wavelet function was used to pre-process the data for two ANN and SVM methods. In addition to the above issues, despite the high potential for solar energy production in Iraq, studies have yet to be conducted in this field. Therefore, investigating the amount of solar energy in Iraq at two different stations using three machine learning methods is one of the first studies in the region mentioned.

Selecting the input variables to evaluate Rs is vital for more accuracy in the model, so it was tried to assess the intensity of solar radiation using the variables that can be measured at the stations. One reason for this act is to prevent errors caused by the measurement tools of input variables, and the other is to increase network performance speed in the training and learning stages. To achieve this purpose, the following plans are followed in the current study: (1) Determining the appropriate number of neurons to obtain the optimal structure in the ANN method. (2) Choosing the most appropriate membership functions in ANFIS. (3) Defining the optimal value of the triple coefficients of the SVM method. (4) Calibration and Validation of ANN, ANFIS, and SVM methods in the estimation of Rs in the selected areas.

## 2. Materials and Methods

### 2.1. Study Area

This research selected two stations with a hot and desert climate in the Wasit and Dhi Qar provinces of Iraq. Figure 1 shows the geographical situation of the two provinces. In general, the number of sunny days in Iraq is high, so the potential of solar energy in this region is estimated to be high. On average, the number of sunny days in the Wasit region is 307 days, and in the Dhi Qar region, it is 312 days. The geographical coordinates of the middle region are 32°40′ N 45°45′ E. Additionally, Dhi Qar is at coordinates 31°02′ N 46°15′ E. The data used in this research are: maximum (Tmax), minimum (Tmin), average temperature (Tavg), average humidity (H avg), and sunshine hours (S) as input and solar radiation (Rs) as the output of the models. It was obtained from the daily time scale from the study stations during the statistical period of 2008–2020 from the Iraqi Agricultural and Meteorological Center (IAC) [19]. Table 1 illustrates the climatic averages of the variables used during this statistical period.

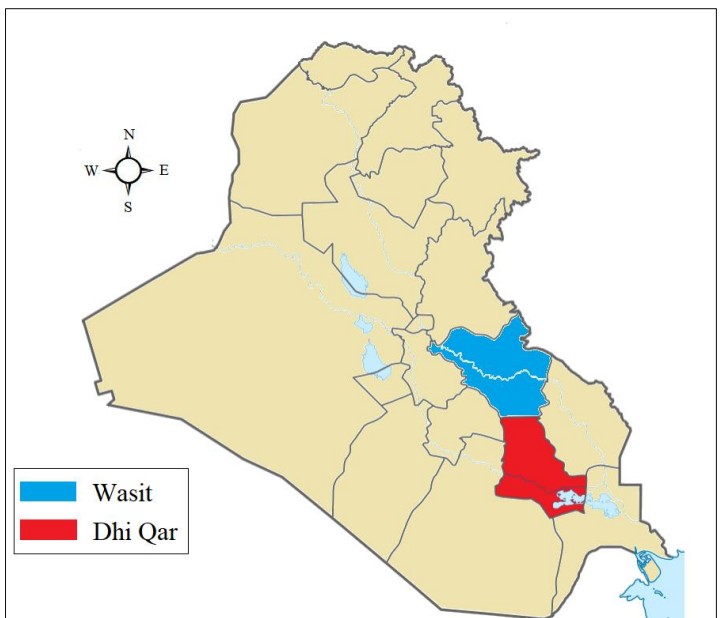

**Figure 1.** The geographical location of Wasit and Dhi Qar provinces.

**Table 1.** Average meteorological data used in the studied stations.

| Station | Wasit | | | Dhi Qar | | |
|---|---|---|---|---|---|---|
| | **Max** | **Min** | **Average** | **Max** | **Min** | **Average** |
| T max (°C) | 43.5 | 17.8 | 31.2 | 44.8 | 17.3 | 32.4 |
| T min (°C) | 25.4 | 5.7 | 15.6 | 28.1 | 6.2 | 17.1 |
| H avg (%) | 95.6 | 3.7 | 28.4 | 97.2 | 5.4 | 29.2 |
| S (h) | 13.8 | 0.0 | 9.7 | 13.4 | 0.0 | 9.2 |
| Rs (MJ/m$^2$ d) | 41.7 | 3.1 | 22.8 | 40.2 | 2.7 | 21.6 |

*2.2. Artificial Neural Network (ANN)*

Artificial neural networks can identify a nonlinear dynamic plan without having defaults in the strategy of modeling. The architecture of an ANN involves a 3-section design, which contains input, hidden, and output sections. The input section prepares the input information to go to the hidden section. Then, before sending information to the output section, the hidden section computes weight coefficients employing transfer functions such as logical functions. The 3-section structure of the ANN is founded on a linear variety of input variables that is changed by a nonlinear function. This design of the ANN is called the leading network. Additionally, within a section, neurons do not communicate with each other. In the ANN, the output of the Y network is modeled as Equation (1) [20–22]:

$$y_i = \sum_{j=1}^{n} W_{i,j} \, X_{i,j} + \theta_i \tag{1}$$

where, $x_{i,j}$ is the input signal from the *j* th neuron in the input section, $W_{i,j}$ is the connection weight of neuron *j* to neuron *i*, and $\theta_i$ is the bias of neuron *i*. During the training process, these weights and fixed values that are added to them and are called bias are changed successively so that the error reaches its lowest value. After calculating each $y_i$, the network starts to modify it under a function. The function is typically a bounded uniform function like the standard sigmoid function and is defined as Equation (2):

$$f_x = \frac{1}{e^{-x} + 1} \tag{2}$$

*2.3. Support Vector Machine (SVM)*

SVM is a method employed to classify and examine the relationship between data. If the data is discrete, SVM leads linear machines to create an optimal surface that divides the data without error and with the highest distance between the surface and the closest training points of the support vectors. If the training points are defined as $[x_i, y_i]$ and the input vector $x_i \in R^n$ and the class value $y_i \in \{-1,1\}$, $i = 1, \ldots$ I, then in the case that the data are linearly separable. The decision rules are defined according to Equation (3) in such a way that they separate the binary decision categories by an optimal surface [23,24]:

$$y = sign \left( \sum_{i=1}^{N} y_i \ a_i \ (X \cdot X_i) + b \right) \tag{3}$$

where $y$ is the output of the equation, and $y_i$ is the category value of the training example and illustrates the inner multiplication. The vector $x = (x_1, x_2, \ldots x_n)$ represents an input datum and the vectors $i = 1, \ldots, N$, $X_i$ are the support vectors. In the equation, the parameters $a$ and $b$ determine the superplane. If the data are not linearly separable, Equation (3) is changed to Equation (4):

$$y = sign \left( \sum_{i=1}^{N} y_i \ a_i \ K(X \cdot X_i) \right) + b \tag{4}$$

where the $K(X \cdot X_i)$ function is a kernel function that generates internal multiplications to create machines with different types of non-linear decision levels in the data space.

*2.4. Wavelet Transform*

Wavelet transform is an efficient mathematical transformation in signal processing. Mathematical transformations are utilized to acquire additional data from the signal that is not available. Wavelet analysis, like Fourier examination, deals with the expansion of functions, but this expansion is completed in terms of wavelets. The wavelet is a supposed characteristic function with zero average. Unlike trigonometric polynomials, they are checked locally in space. In this way, a closer relationship between some functions and their coefficients is possible, and more numerical stability is supplied in reconstruction and measures. Any application founded on a fast Fourier analysis can be formulated employing wavelets and obtaining more local spatial data [25,26].

The wavelet has two essential characteristics, short-term and volatility. In other words, $\psi(x)$ is a wavelet function if and only if its Fourier transform $\psi(\omega)$ satisfies the following situation [26]:

$$\int_{-\infty}^{+\infty} \frac{|\psi(\omega)|}{|\omega|^2} \, d\omega < +\infty \tag{5}$$

This condition is an acceptance condition of the $\psi(x)$ wavelet. The above equation can be considered equivalent to the following formula that must be satisfied [26]:

$$\psi(0) = \int_{-\infty}^{+\infty} \psi(x) \, dx = 0 \tag{6}$$

This function feature with zero mean is not limiting, and numerous functions can be named wavelet functions according to it. $\psi(x)$ is the main wavelet function that the functions employed in the investigation transfer and scaling changes the size and location during the signal analysis, and in the end, the wavelet coefficients for each value of the scale ($a$) and at each point of the signal ($b$) can be calculated with Equation (7) [26]:

$$\psi_{a,b}(x) = \frac{1}{\sqrt{a}} \, \psi \left( \frac{x - b}{a} \right) \tag{7}$$

As a mathematical operator, the act of scaling expands or compresses the signal. For the assumed function $f(t)$, if $s < 1$, $f(st)$ is the expanded state of $f(t)$, and if $s > 1$, it is the compressed state of $f(t)$. As can be seen in Equation (7), the term a is in the denominator and, therefore, is the opposite of what was said; if $a < 1$, the signal is compressed, and if $a > 1$, the signal is expanded. Additionally, in the above relationship, the parameter b models delay and precedence on a function.

### 2.5. Adaptive Neural Fuzzy Inference System (ANFIS)

ANFIS is a multilayer feed-forward approach that uses fuzzy logic and neural network learning algorithms to map an input space to an output space. The structure of this model involves 5 layers of input nodes, base nodes, intermediate nodes, result nodes, and output nodes. If the result of every neuro-fuzzy network layer is exhibited as $Q_{i,j}$, which is the output of the i-th group in the 1st node, the execution of the different layers can be expressed as follows [27,28]:

Layer 1: Every node in this layer is equal to a fuzzy set, Equation (8).
Layer 2: The input signals are multiplied and create output, Equation (9).
Layer 3: This layer calculates the proportion of the activity grade of the i-th rule to the sum of the activity degrees of all the rules, Equation (10).
Layer 4: The result of every node in this layer is as seen in Equation (11).
Layer 5: Each node in this layer, which is displayed as $\sum$, calculates the last output value in the form of Equation (12).

$$Q_i^1 = \mu_{Ai}(x) \tag{8}$$

$$w_i = \mu_{Ai}(x) \times \mu_{Bi}(y) \qquad i = 1, 2 \tag{9}$$

$$\overline{W}_i = \frac{w_i}{w_1 + w_2}, \qquad i = 1, 2 \tag{10}$$

$$Q_i^4 = \overline{W}_i f_i = \overline{W}_i(p_i x + q_i y + r_i) \tag{11}$$

$$Q_i^5 = \sum_i \overline{W}_i f_i = \frac{\sum_i W_i f_i}{\sum_i W_i} \tag{12}$$

where $\overline{W}_i$ is the output of the third layer and $\{p_i, q_i, r_i\}$ are the set of adaptive parameters of this layer. These parameters are called result parameters.

### 2.6. Criteria for Evaluating

To evaluate and compare the accuracy of the used methods, root mean square error (RMSE), mean absolute error (MAE), index of agreement (IA), and the coefficient of determination ($R^2$) were used.

RMSE is the difference between the value predicted by the model and the actual value and is a good tool for comparing the prediction errors of a data set (Equation (13)) [29]:

$$\text{RMSE} = \sqrt{\frac{\sum_{i=1}^n (O_i - P_i)^2}{n}} \tag{13}$$

$R^2$ (Equation (14)) indicates how much of the changes in the dependent variable are influenced by the corresponding independent variable, and the rest of the dependent variables are related to other factors. $R^2$ is between zero and one. Zero indicates that the model does not determine any of the variability of the response data around its average. However, a value of one indicates that all observed values will be the same as the fitted values, and all the data points will lie on the fitted line [29].

$$R^2 = \sqrt{\frac{\left[\sum_{i=1}^n (O_i - \overline{O})(P_i - \overline{P})\right]^2}{\sum_{i=1}^n (O_i - \overline{O})^2 (P_i - \overline{P})^2}} \tag{14}$$

MAE, ideally, should be zero, and negative and positive values indicate underestimation and overestimation, respectively. The Equation (15) of this parameter represents the method's accuracy and the error's average value [29].

$$\text{MAE} = \frac{\sum_{i=1}^{n}(O_i - P_i)^2}{n} \tag{15}$$

AI has a value between zero and one, where a value of one represents the best fit and is calculated according to Equation (16) [29].

$$\text{AI} = \frac{\sum_{i=1}^{n}|P_i - O_i|}{\sum_{i=1}^{n}\left|\left(P_i - \overline{P}\right)^2\right| + \left|\left(O_i - \overline{O}\right)^2\right|} \tag{16}$$

In Equations (13)–(16), $P_i$ is the predicted radiation intensity, $\overline{P}$ is the mean predicted radiation intensity, $O_i$ is the measured radiation intensity, $\overline{O}$ is the mean measured radiation intensity, and $n$ is the number of recorded data.

### 2.7. Training, Validation, and Testing Sets

The approach of soft data computing models is based on the use of criteria for dividing data into training, validation, and test sets to create and evaluate the performance of these algorithms. The training, validation, and test sets are defined as follows [30]:

A training set is a set of examples that are used to develop learning training and create a soft computing base data model. The case data sample is used to fit the model. The validation set is a subset of the training set, which includes several examples to adjust model parameters and control the training process. The validation data set can be used to stop training the model. The test set is a set of samples used only to evaluate and correct the performance of the created base data model. In this study, 70%, 15%, and 15% of the data were used for the training, validation, and test sets, respectively.

### 3. Results

For Wasit and Dhi Qar station, among the selected models, WANN-2 with log-sigmoid function and arrangement (4-1-25) in Dhi Qar station and WANN-3 with log-sigmoid function and arrangement (4-1-20) in Wasit station were the best models. After 38 runs, the desired optimal parameters algorithm was obtained for the WSVM method. For the SWVM method, at Dhi Qar station, the optimal parameters were WSVM-3 with C = 500, $\gamma$ = 5, and $\varepsilon$ = 0.06 for the error parameter. For Wasit station, WSVM-1 with C = 500, $\gamma$ = 15, and $\varepsilon$ = 0.07 was chosen as the best model. Additionally, for each station, among the selected models, ANFI-2 with membership functions Trimf in Dhi Qar station and ANFIS-4 with membership functions Pimf in Wasit station were the best models.

To assess and compare the performance of models with each other, training and test data were used. The output values of each method were compared with the corresponding observed values, and its details were analyzed based on error measurement statistics. The results of the optimal structures of WANN, WSVM, and ANFIS in estimating the value of Rs separately for each station are shown in Tables 2–4. Comparing the performance of WANN, WSVM, and ANFIS with error measurement criteria (RMSE, $R^2$, MEA and IA) demonstrated that these models were able to predict solar energy.

**Table 2.** The results of statistical analysis of different input patterns of WANN model in Rs evaluation.

| Station | | Dhi Qar | | | | Wasit | | | |
|---|---|---|---|---|---|---|---|---|---|
| Model | | WANN-1 | WANN-2 | WANN-3 | WANN-4 | WANN-1 | WANN-2 | WANN-3 | WANN-4 |
| **Training phase** | RMSE | 3.10 | 2.76 | 3.00 | 3.08 | 3.01 | 3.40 | 2.42 | 3.21 |
| | MEA | 2.24 | 2.09 | 2.20 | 2.18 | 2.02 | 2.21 | 1.80 | 2.07 |
| | $R^2$ | 0.82 | 0.86 | 0.77 | 0.78 | 0.82 | 0.76 | 0.89 | 0.79 |
| | IA | 0.96 | 0.97 | 0.96 | 0.95 | 0.95 | 0.95 | 0.98 | 0.96 |
| **Validation Phase** | RMSE | 3.18 | 2.83 | 3.20 | 3.36 | 3.08 | 3.38 | 2.59 | 3.22 |
| | MEA | 2.28 | 2.07 | 2.28 | 2.38 | 2.15 | 2.21 | 2.04 | 2.13 |
| | $R^2$ | 0.80 | 0.85 | 0.77 | 0.76 | 0.79 | 0.77 | 0.88 | 0.74 |
| | IA | 0.96 | 0.95 | 0.95 | 0.93 | 0.93 | 0.94 | 0.96 | 0.95 |
| **Testing Phase** | RMSE | 3.39 | 2.93 | 3.24 | 3.47 | 3.1 | 3.30 | 2.84 | 3.20 |
| | MEA | 2.33 | 2.09 | 2.31 | 2.43 | 2.14 | 2.23 | 2.05 | 2.14 |
| | $R^2$ | 0.79 | 0.85 | 0.76 | 0.74 | 0.80 | 0.79 | 0.88 | 0.76 |
| | IA | 0.94 | 0.96 | 0.93 | 0.93 | 0.95 | 0.95 | 0.97 | 0.96 |

**Table 3.** The results of statistical analysis of different input patterns of WSVM model in Rs evaluation.

| Station | | Dhi Qar | | | | Wasit | | | |
|---|---|---|---|---|---|---|---|---|---|
| Model | | WSVM-1 | WSVM-2 | WSVM-3 | WSVM-4 | WSVM-1 | WSVM-2 | WSVM-3 | WSVM-4 |
| **Training phase** | RMSE | 2.88 | 3.06 | 2.45 | 3.13 | 2.67 | 3.02 | 3.00 | 2.69 |
| | MEA | 2.04 | 2.08 | 1.92 | 2.12 | 1.58 | 1.91 | 1.90 | 1.84 |
| | $R^2$ | 0.82 | 0.79 | 0.87 | 0.78 | 0.88 | 0.79 | 0.80 | 0.86 |
| | IA | 0.96 | 0.96 | 0.97 | 0.95 | 0.97 | 0.96 | 0.96 | 0.96 |
| **Validation Phase** | RMSE | 3.07 | 3.57 | 2.49 | 3.52 | 2.45 | 3.86 | 3.38 | 2.99 |
| | MEA | 2.18 | 2.19 | 2.11 | 2.21 | 1.67 | 3.84 | 3.19 | 1.98 |
| | $R^2$ | 0.75 | 0.66 | 0.85 | 0.74 | 0.86 | 0.77 | 0.88 | 0.82 |
| | IA | 0.94 | 0.94 | 0.96 | 0.94 | 0.93 | 0.95 | 0.94 | 0.95 |
| **Testing Phase** | RMSE | 3.37 | 3.77 | 2.54 | 3.68 | 2.45 | 3.92 | 3.42 | 3.27 |
| | MEA | 2.22 | 2.37 | 2.18 | 2.30 | 1.81 | 2.22 | 2.06 | 2.21 |
| | $R^2$ | 0.74 | 0.67 | 0.85 | 0.69 | 0.86 | 0.72 | 0.76 | 0.81 |
| | IA | 0.95 | 0.92 | 0.97 | 0.93 | 0.95 | 0.93 | 0.94 | 0.96 |

**Table 4.** The results of statistical analysis of different input patterns of ANFIS model in Rs evaluation.

| Station | | Dhi Qar | | | | Wasit | | | |
|---|---|---|---|---|---|---|---|---|---|
| Model | | ANFIS-1 | ANFIS-2 | ANFIS-3 | ANFIS-4 | ANFIS-1 | ANFIS-2 | ANFIS-3 | ANFIS-4 |
| **Training phase** | RMSE | 2.94 | 2.62 | 2.85 | 2.92 | 2.86 | 3.23 | 3.04 | 2.70 |
| | MEA | 2.12 | 1.98 | 2.09 | 2.07 | 1.91 | 2.10 | 1.96 | 1.91 |
| | $R^2$ | 0.74 | 0.83 | 0.73 | 0.77 | 0.80 | 0.72 | 0.75 | 0.84 |
| | IA | 0.91 | 0.94 | 0.93 | 0.90 | 0.89 | 0.92 | 0.93 | 0.95 |
| **Validation Phase** | RMSE | 3.18 | 2.59 | 2.94 | 3.19 | 2.89 | 3.17 | 3.00 | 2.65 |
| | MEA | 2.14 | 2.08 | 2.11 | 2.24 | 2.00 | 2.09 | 2.17 | 2.10 |
| | $R^2$ | 0.71 | 0.81 | 0.73 | 0.75 | 0.84 | 0.69 | 0.71 | 0.81 |
| | IA | 0.90 | 0.93 | 0.91 | 0.85 | 0.86 | 0.88 | 0.91 | 0.94 |
| **Testing Phase** | RMSE | 3.22 | 2.64 | 3.07 | 3.29 | 2.94 | 3.13 | 3.03 | 2.69 |
| | MEA | 2.21 | 2.13 | 2.19 | 2.31 | 2.03 | 2.11 | 2.15 | 2.11 |
| | $R^2$ | 0.70 | 0.78 | 0.72 | 0.70 | 0.76 | 0.70 | 0.72 | 0.80 |
| | IA | 0.89 | 0.92 | 0.90 | 0.88 | 0.87 | 0.89 | 0.91 | 0.92 |

For Wasit station, according to the results obtained from the WANN, the lowest values of RMSE, MAE, and the highest values of $R^2$ and AI were related to WANN-3. The amount of RMSE, MAE, $R^2$, and AI in the training phase were 3.42, 1.8, 0.89, and 0.98, respectively, and for the testing phase, the amounts were 2.84, 2.05, 0.88, and 0.97. However, the results of WANN-1 for Wasit station were also somewhat acceptable, but the difference with WANN-3 is significant. In the WANN model, the results obtained for Dhi Qar station from the WANN-2 model had the best values for the RMSE, MEA, $R^2$, and AI criteria. The criteria values were equal to 2.93, 2.09, 0.86, and 0.97 in the training phase and 2.78, 2.09, 0.85, and 0.96 in the test phase. The investigation revealed that the answers of the WANN-1 model at Dhi Qar station were also acceptable.

For Wasit station, based on the results obtained from the WSVM, the lowest values of RMSE, MAE, and the highest values of $R^2$ and AI were related to WSVM-1. The amount of RMSE, MAE, $R^2$, and AI in the training phase were 2.67, 1.58, 0.88, and 0.97, respectively, and for the testing phase, the amount was 2.45, 1.81, 0.86, and 0.95. However, the WSVM-4 model at Wasit station had close results with the WSVM-1 model, which was the best model at this station. The results of the other two models, WSVM-2 and WSVM-3, were acceptable. In the WSVM model, the results obtained for Dhi Qar station from the WANN-3 model had the best values for the RMSE, MEA, $R^2$, and AI criteria. The criteria values were equal to 2.45, 1.92, 0.87, and 0.97 in the training phase and 2.54, 2.18, 0.85, and 0.97 in the test phase. In this station also, the WSVM-1 model obtained good results.

For Wasit station, according to the results obtained from the ANFIS, the best results of RMSE, MAE, $R^2$, and AI were related to ANFIS-4. The amount of RMSE, MAE, $R^2$, and AI in the training phase were 2.7, 1.91, 0.84, and 0.92, respectively, and for the testing phase, the amount was 2.69, 2.11, 0.8, and 0.92. The results of the ANFIS-1 model in this station were also good. In the ANFIS model, the results obtained for Dhi Qar station from the ANFIS-2 model had the best values for the RMSE, MEA, $R^2$, and AI criteria. The criteria values were 2.62, 2.13, 0.83, and 0.94 in the training phase and 2.64, 1.98, 0.78, and 0.92 in the test phase. The results of other models for this station differed greatly from the best model.

The correlation between the Rs values evaluated by WANN, WSVM, and ANFIS models and the observed values of each station are shown in Figures 2 and 3 separately from the two training and testing stages. This result showed the relationship between the observed Rs values and estimated values from the WANN, WSVM, and ANFIS models. The figures shown for each model (WANN, WSVM, and ANFIS) were taken from the best model of each method. This was WANN-3 and WANN-2 for the WANN model in Wasit and Dhi Qar stations, respectively, and WSVM-1 and WSVM-3 for the WSVM model in Wasit and Dhi Qar stations, respectively. Finally, ANFIS-2 and ANFIS-4 were used for the ANFIS model in Wasit and Dhi Qar stations, respectively.

As shown in Figures 2 and 3, both WAAN and WSVM models had higher $R^2$ values in both the training and the test phases compared to the ANFIS model in both Wasit and Dhi Qar stations. These results showed that the performance of the WANN and WSVM models was better than the ANFIS model. However, the ANFIS model recorded values higher than 0.78 in both stations for the training and testing phases, which showed that this model could also record acceptable estimation. However, if it is possible to use two other models, it is better to use the WAAN and WSVM models.

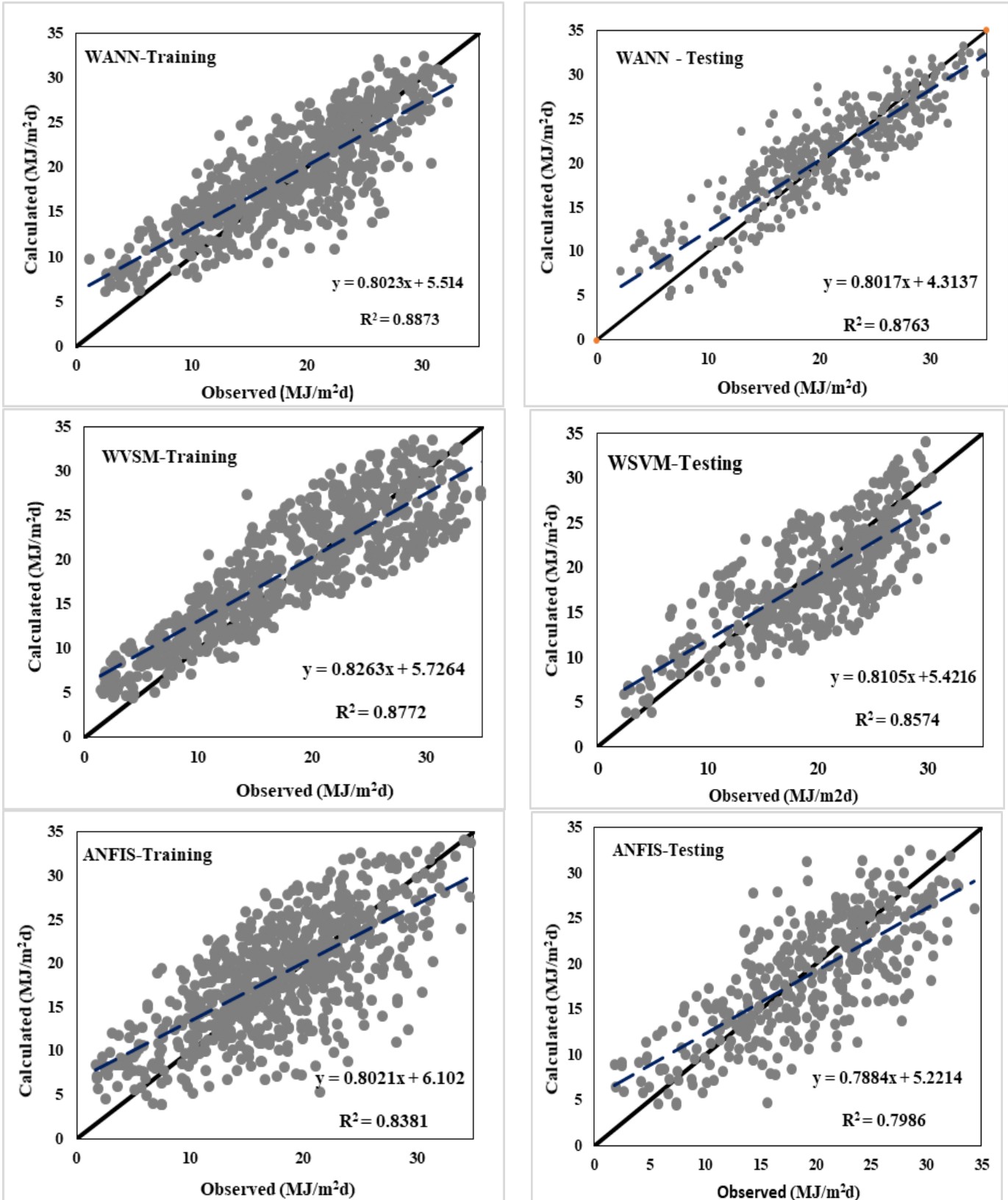

**Figure 2.** Correlation between radiation intensity (Rs) assessed by WANN, WSVM, and ANFIS methods and observed values at Wasit station in two stages of training and testing.

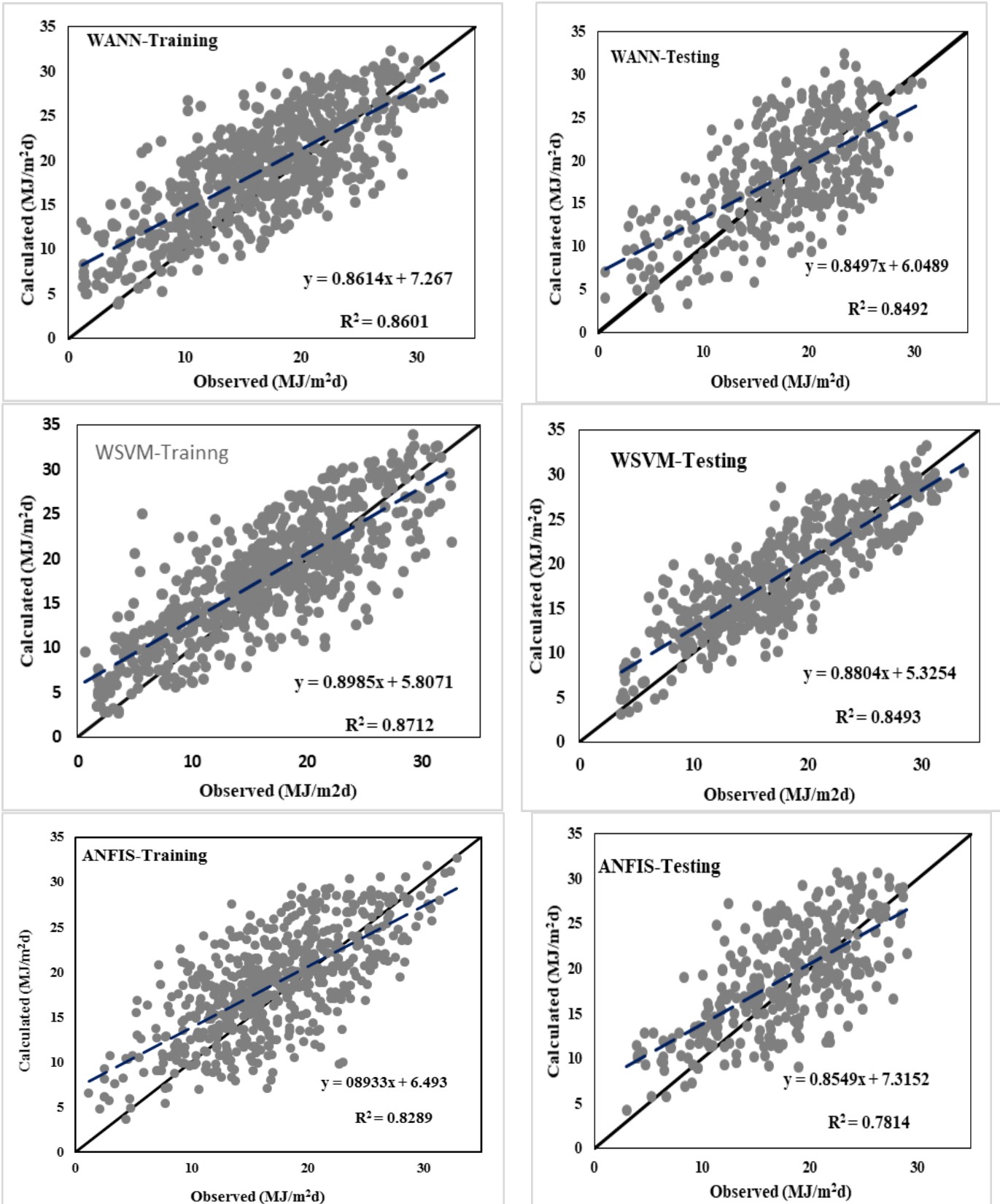

**Figure 3.** Correlation between radiation intensity (Rs) assessed by WANN, WSVM, and ANFIS methods and observed values at Dhi Qar station in two stages of training and testing.

## 4. Discussion

Today, the estimation and forecasting of solar energy have been developed using artificial intelligence and machine learning models. This study aimed to evaluate the effectiveness of the WANN, WSVM, and ANFIS methods in estimating solar energy in Wasit and Dhi Qar stations. This research used the wavelet for data preprocessing of the ANN and SVM models. In order to evaluate solar energy in two stations, data related to the maximum temperature, minimum temperature, average humidity, and sunny hours were considered as input to the models. Using statistical indicators for the WANN method, RSME = 2.42, EMA = 1.8, $R^2$ = 0.89, and IA = 0.98 was obtained in Wasit station, and RSME = 2.84, EMA = 2.05, $R^2$ = 0.86, and IA = 0.97 was obtained in Dhi Qar station. In the WSVM method, RSME = 2.67, EMA = 1.58, $R^2$ = 0.88, and IA = 0.97 for Wasit station. In Dhi Qar station, RSME = 2.54, EMA = 1.92, $R^2$ = 0.87, and IA = 0.97. In the ANFIS method, for Wasit station, RSME = 2.7, EMA = 1.91, $R^2$ = 0.84, and IA = 0.95. In Dhi Qar station, RSME = 2.62, EMA = 1.98, $R^2$ = 0.83, and IA = 0.94. The results of this study indicated that all three used models predict the results well. WANN and WSVM methods in solar energy modeling had similar results. However, the results of the WANN model were slightly better than the WSVM model. The ANFIS method also obtained acceptable results but compared to the other two models, ANFIS results were less accurate. The better results of WANN and WSVR could be due to the use of the wavelet model in data preprocessing. The results of this research are similar to research conducted in Mexico. In the research conducted in Mexico, two ANN and SVM models had better results than an ANFIS model [31]. Additionally, a study completed in Nigeria showed that the SVM model had better results than the ANFIS model [14]. In this research, the MLP model was used for the learning process in the WANN method. It is suggested to use EFB and CFB methods in future studies and evaluate the best learning method. Additionally, using the Gene Expression Programming (GEP) black box model to estimate radiation intensity and compare it with other models can be a subject for researchers in future studies.

## 5. Conclusions

With the increase in the need for energy around the world, clean energy, such as solar energy, is receiving more attention. Investigating the potential of solar energy using physical tools is difficult and expensive. This research showed that using machine learning methods is suitable for estimating the amount of solar energy. In this research, it was demonstrated that WANN, WSVM, and ANFIS models have an excellent ability to estimate the amount of solar energy. In Wasit and Dhi Qar stations, the value of $R^2$ for the WANN and WSVM methods was 0.89 and 0.86, respectively. The value of $R^2$ in the WANN and WSVM methods in the Wasit and Dhi Qar stations was 0.88 and 0.87, respectively. The ANFIS technique also obtained acceptable results. However, compared to the other two techniques, the ANFIS results were lower, and the $R^2$ value was 0.84 and 0.83 in Wasit and Dhi Qar stations, respectively.

**Author Contributions:** Conceptualization, W.A.; Methodology, S.A. and A.A.-M.; Software, W.A.; Validation, W.A.; Formal analysis, M.J.J. and S.A.; Investigation, M.J.J. and Y.C.-E.; Resources, A.A.-M. and Y.C.-E.; Data curation, I.H.A.-K. and A.A.-M.; Writing—original draft, I.H.A.-K.; Writing—review & editing, Y.C.-E. All authors have read and agreed to the published version of the manuscript.

**Funding:** This research received no external funding.

**Data Availability Statement:** Not applicable.

**Conflicts of Interest:** The authors declare no conflict of interest.

## Abbreviations

| Symbols | definition | Unit |
|---|---|---|
| ANN | Artificial Neural Network | - |
| SVM | Support Vector Machine | - |
| ANFIS | Adaptive Neuro-Fuzzy Inference System | - |
| WANN | Wavelet Artificial Neural Network | - |
| WSVM | Wavelet Support Vector Machine | - |
| Rs | solar radiation | $MJ/m^2$ d |
| TWh | Terawatt-hour | - |
| GIS | Geographic Information System | - |
| T max | Maximum Temperature | $°C$ |
| T min | Minimum Temperature | $°C$ |
| H ave | Average Humidity | % |
| T avg | Average Temperature | $°C$ |
| $W_{i,j}$ | The Connection Weight of Neuron $j$ to Neuron $i$ | - |
| $K(X.X_i)$ | a Kernel Function | - |
| $\theta_i$ | The Bias of Neuron $i$ | - |
| $\psi(x)$ | Wavelet Function | - |
| $\overline{W}_i$ | The Output of The Third layer in ANFIS Model | - |
| $\{p_i,\,q_i,\,r_i\}$ | Set of Adaptive parameters | - |
| RMSE | Root Mean Square Error | - |
| MAE | Mean Absolute Error | - |
| IA | Index of Agreement | - |
| $R^2$ | Coefficient of Determination | - |
| $P_i$ | The Predicted Radiation Intensity | $MJ/m^2$ d |
| $\overline{P}$ | The Mean Predicted Radiation Intensity | $MJ/m^2$ d |
| $O_i$ | The Measured Radiation Intensity | $MJ/m^2$ d |
| $\overline{O}$ | The Mean Measured Radiation Intensity | $MJ/m^2$ d |
| $n$ | The Number of Recorded Data | - |

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
