# Peer review of "Comparison of Wavelet Artificial Neural Network, Wavelet Support Vector Machine, and Adaptive Neuro-Fuzzy Inference System Methods in Estimating Total Solar Radiation in Iraq"

_energies, doi:10.3390/en16020985_

Round 1
Reviewer 1 Report
This manuscript can be accepted for publication after major revisions, see the followings:
*English should be improved.
*The Abstract should be improved.
*Better description and explanation on the figures (Figures 2 to 3).
*Better description and explanation on the tables (Figures 2 to 4).
*The validation part should be added.
* The novelty is not clear.
*The Conclusion could, also, be improved.
* Introduction part needs to be extended by some of the recently published papers to show the importance of Solar Energy Systems in high-quality journals. The following references should be included in this manuscript:
[1] Parimita Panigrahi, S., Kumar Maharana, S., Rajashekaraiah, T., Gopalashetty, R., Sharifpur, M., Ahmadi, M. H., ... & Abbas, M. (2022). Flat Unglazed Transpired Solar Collector: Performance Probability Prediction Approach Using Monte Carlo Simulation Technique. Energies, 15(23), 8843.
[2] Sharifpur, M., Ahmadi, M. H., Rungamornrat, J., & Malek Mohsen, F. (2022). Thermal Management of Solar Photovoltaic Cell by Using Single Walled Carbon Nanotube (SWCNT)/Water: Numerical Simulation and Sensitivity Analysis. Sustainability, 14(18), 11523.
*I hope that the authors refer to more published papers in the Energies.
Author Response
Thank you for your valuable comments. Please find the response file in the attachment.

Reviewer 2 Report
The manuscript is interesting and provide useful data; however, it needs following revisions to become acceptable for publication:
1. Adding some important quantitative data and results in abstract is suggested.
2. Writing needs recheck and improvement.
3. Since the study is on solar radiation, it is better to consider "radiation" in the title of the manuscript.
4. Unit of data must e added on both vertical and horizontal axes of the figures, i.g. figure 2.
5. In conclusion section, it is better to avoid representing several quantitative results, it is better to just indicate the most important ones.
6. Adding nomenclature and definition of abbreviations is suggested.
7. Following references are suggested to improve literature review on the applications of intelligent methods and solar energy systems:
"Thermophysical Properties of Hybrid Nanofluids and the Proposed Models: An Updated Comprehensive Study" https://doi.org/10.3390/nano11113084
"PV/Thermal as Promising Technologies in Buildings: A Comprehensive Review on Exergy Analysis" https://doi.org/10.3390/su141912298
"Estimating Solar Energy Potential in Eritrea: a GIS-based Approach" https://doi.org/10.22044/rera.2022.11737.1106
Author Response

(The authors gave the same response as above.)

Reviewer 3 Report
My suggestions are below.
The introduction should be enriched with references.
It should explain the study area in detail.
The conclusion part should be added.
Author Response

(The authors gave the same response as above.)

Round 2
Reviewer 1 Report
This article can be accepted.
Reviewer 3 Report
Accept